# Measurement report: Variations in surface $SO_2$ and $NO_x$ mixing ratios from 2004 to 2016 at a background site in the North China Plain

Xueli Liu[1], Liang Ran[2], Weili Lin[1*], Xiaobin Xu[3], Zhiqiang Ma[4], Fan Dong[4], Di He[4], Liyan Zhou[4], Qingfeng Shi[4], and Yao Wang[4]

[1] Key Laboratory of Ecology and Environment in Minority Areas (Minzu University of China), National Ethnic Affairs Commission, Beijing 100081, China

[2] Key Laboratory of Middle Atmosphere and Global Environment Observation, Institute of Atmospheric Physics, Chinese Academy of Sciences, Beijing, 100089, China

[3] Chinese Academy of Meteorological Sciences, Beijing, 100081, China

[4] Beijing Shangdianzi Regional Atmosphere Watch Station, Beijing, 101507, China

*Correspondence to*: Weili Lin (linwl@muc.edu.cn)

**Abstract.** Strict air pollution control strategies have been implemented in recent decades in the North China Plain
(NCP), previously one of the most polluted regions in the world, and have resulted in considerable changes in
emissions of air pollutants. However, little is so far known about the long-term trends of the regional background
level of $NO_x$ and $SO_2$, along with the increase and decrease processes of regional emissions. In this study, the
seasonal and diurnal variations of $NO_x$ and $SO_2$ as well as their long-term trends at a regional background station in
the NCP were characterized from 2004 to 2016. On average, $SO_2$ and $NO_x$ mixing ratios were 5.7 ± 8.4 ppb and
14.2 ± 12.4 ppb, respectively. The seasonal variations in $SO_2$ and $NO_x$ mixing ratios showed a similar pattern with a
peak in winter and a valley in summer. However, the diurnal variations in $SO_2$ and $NO_x$ mixing ratios greatly
differed for all seasons, indicating different sources for $SO_2$ and $NO_x$ and meteorological effects on their
concentrations. Overall, the annual mean $SO_2$ exhibited a significant decreasing trend of $-6.1$ % $yr^{-1}$ ($R = -0.84$, $P < $
0.01) from 2004 to 2016, which is very close to $-6.3$ % $yr^{-1}$ of the annual $SO_2$ emission in Beijing, and a greater
decreasing trend of $-7.4$ % $yr^{-1}$ ($R = -0.95$, $P < 0.01$) from 2008 to 2016. The annual mean of $NO_x$ showed a
fluctuating rise of $+3.4$ % $yr^{-1}$ ($R = 0.38$, $P = 0.40$) from 2005 to 2010, reaching the peak value (16.9 ppb) in 2010,
and then exhibited an extremely significant fluctuating downward trend of $-4.5$ % $yr^{-1}$ ($R = 0.95$, $P < 0.01$) from
2010 to 2016. After 2010, the annual mean $NO_x$ mixing ratios correlated significantly ($R = 0.94$, $P < 0.01$) with the
annual $NO_x$ emission in North China. The decreasing rate ($-4.8$ % $yr^{-1}$, $R = -0.92$, $P < 0.01$) of the annual mean
$NO_x$ mixing ratios from 2011 to 2016 at the Shangdianzi (SDZ) regional atmospheric background station are lower
than the one ($-8.8$ % $yr^{-1}$, $R = -0.94$, $P < 0.01$) for the annual $NO_x$ emission in the NCP and ($-9.0$ % $yr^{-1}$, $R = -0.96$,
$P < 0.01$) in Beijing. It indicated that surface $NO_x$ mixing ratios at SDZ had weaker influence than $SO_2$ by the
emission reduction in Beijing and its surrounding areas in the NCP. The increase in the amount of motor vehicles
led to an increase in traffic emissions for $NO_x$. This study supported conclusions from previous studies that the
measures taken for controlling $NO_x$ and $SO_2$ in the NCP in the past decades were generally successful. However,
$NO_x$ emission control should be strengthened in the future.
**1 Introduction**
Acid gases sulfur dioxide ($SO_2$) and nitrogen oxides ($NO_x$) are closely related to climate, ecology, environment and
human health. They are important gaseous pollutants in China (Xu et al., 2009) and also recommended by the
Global Atmosphere Watch (GAW) of the World Meteorological Organization (WMO) for priority observation
(WMO, 2001). They can also be transformed into nitrate and sulfate aerosols, which play an important role in the
formation of aerosol pollution and acid rain (Yang et al., 2011; Cheng et al., 2013; Luo et al., 2016; Chen et al.,
2017a). Sulfate and nitrate constitute more than 1/3 of $PM_{2.5}$ mass concentration and can cause serious respiratory
diseases (Yang et al., 2010; Yang et al., 2011; Gao et al., 2012; Zhao et al., 2013; Liu et al., 2014a).
With the economic development, population growth and rapid urbanization, air pollution in China exhibited the
characteristics of regional pollution centering urban areas in recent years (Shao et al., 2006; Xu et al., 2008). Many
studies thereby focused on regional pollution (Qi et al., 2012; Li et al., 2015), instead of local and suburban
pollution as previously did (Liu et al., 2008; Lin et al., 2009a). Local/suburban pollution is closely associated air
pollutants emitted locally and limited on a smaller scale such as a town, a city or an urban area. Regional pollution
occurs over the whole region and is usually associated with large-scale emissions and significantly influenced by
transport and accompanying processes, such as chemical reactions, deposition, etc. In China, city clusters have been
formed for decades, air pollution often shows regional characteristics. Especially, the North China Plain (NCP)
region, a heavily industrialized and densely populated area with considerable agricultural activities, is one of the
most polluted regions of the world. The strong emissions of $SO_2$ and $NO_x$ in the NCP showed the typical regional
characteristics (Wu et al., 2010; Lin et al., 2012; Liu et al., 2014b), i.e., similar changes in seasonal and diurnal
patterns of $NO_x$ and $SO_2$ had been observed at different types of sites in this region in previous studies. Previous
studies have combined observations at the background site and the urban site for comparisons (Liu et al., 2014b), or
selected short-term observations (1–2 years or 1–2 seasons) for the comparative study before and after major
activities, in order to quantitatively evaluate the effect of implementing control measures during the event (Cheng et
al., 2015; Li et al., 2019; Lin et al., 2011a; Lin et al., 2012; Wei et al., 2016; Wu et al., 2010; Zhong et al., 2020).
Most of the long-term studies (more than 10 years) evaluated the temporal and spatial variations of $SO_2$ and $NO_x$
based on satellite measurements of the vertical column density (Zhang et al., 2007; Cai et al., 2018; Shikwambana et
al., 2020). However, there were few studies on the long-term trend of $SO_2$ and $NO_x$ based on ground-level
observations (Bai et al., 2015), especially in the background area of the NCP and with a time span of more than 10
years.
In this study, we analyzed the long-term variations in surface $SO_2$ and $NO_x$ mixing ratios observed at a regional
WMO GAW station in the NCP, and discussed their influencing factors and their responses to pollution control
measures, so as to provide scientific basis for designing further strategies for controlling $SO_2$ and $NO_x$ on a regional
scale.
**2 Data and methods**
Surface $SO_2$ and $NO_x$ mixing ratios were measured at the Shangdianzi (SDZ) regional atmospheric background
station (117°07′ E, 40°39′ N, 293.3 m a.s.l.). SDZ is located in Shangdianzi Village in Miyun District of Beijing,
China. It is about 110 km northeast of urban Beijing. The measurements of air pollutants at this site could represent
the background conditions in the NCP (Lin et al., 2008; Meng et al., 2009a).
SDZ is situated on the north hill side of a northeast-to-southwest valley, with farmlands in the south. Corn and
wheat were the main crops, but were recently replaced by fruit trees. It lies in a warm temperate and semi-humid
climate zone, with short spring and autumn but long winter and summer. The monthly averages of meteorological
parameters such as temperature (T), pressure (P), precipitation (PRCP), relative humidity (RH), wind speed (WS)
and wind direction are shown in Fig. 1. Precipitation occurs mainly in summer. The prevailing wind directions were
from NE–ENE and WSW–SW. Stronger wind speeds appear in spring and weaker in summer.
In-situ measurements of $SO_2$ and $NO_x$ mixing ratios were made using a pulsed fluorescence $SO_2$ analyzer (Model
43C-TL, Thermo Fisher Scientific, MA, USA) and a chemiluminescence $NO_x$ analyzer (Model 42C-TL, Thermo
Fisher Scientific, MA, USA), respectively. The detection limits of the Model 43C-TL $SO_2$ analyzer and the Model
42C-TL $NO_x$ analyzer are 0.05 ppb (300 second averaging time) and 0.05 ppb (120 second averaging time),
respectively. In Model 42C-TL $NO_x$ analyzer, $NO_2$ is converted to NO by a molybdenum $NO_2$-to-NO converter
heated to about 325℃. The conversion efficiency was checked annually using gas phase titration of an NO standard
with $O_3$. The converter was replaced if the conversion efficiency was found lower than 96%. The drawback in this

NO$_2$ converter was known to suffer from the interference of other NO$_y$ compounds such as PAN and HNO$_3$ (Steinbacher et al., 2007; Jung et al., 2017), which was also discussed in Yin et al. (2022). As it is not possible in our case to remove the interference, the reported NO$_2$ and NO$_x$ levels should be treated as upper limits. In order to obtain long-term trends of atmospheric components at a regional atmospheric background station, the observations are required to be accurate, reliable, and comparable. Therefore, strict and effective quality control measures were implemented during the observation (Lin et al., 2019). Daily zero and span checks were routinely and automatically carried out. Multi-point calibrations were done monthly. The standard gases used at the site were compared against NIST-traceable standard gases to ensure the data comparability (Lin et al., 2009b). During the period from January 2005 to May 2017, the percentages of effective hourly mean data of SO$_2$ and NO$_x$ are 97.1 % and 96.7 %, respectively. The wind speed (WS), wind direction (WD), air temperature (T), precipitation (PRCP), relative humidity (RH), and atmospheric pressure (P) during the same period are from the routine meteorological observations. We used a hybrid single-particle Lagrangian integrated trajectory model (Hysplit4.9) from National Oceanic and Atmospheric Administration, USA, with the NCEP–NCAR reanalysis meteorological data set (https://ready.arl.noaa.gov/archives.php) to calculate the atmospheric mixed layer heights.

## 3 Results

### 3.1 Observational levels

The time series and statistic results of hourly mean SO$_2$ and NO$_x$ mixing ratios during the observational period at SDZ are showed in Fig. 2 and Table 1, respectively. The hourly mean SO$_2$ mixing ratios ranged from 0.01 to 100.34 ppb, with 193 hours (0.18 %) exceeded the limit of 57 ppb set in China National Ambient Air Quality Standard (GB3095–2012, Grade I). The hourly mean NO$_2$ mixing ratios ranged from 0.01 to 124.4 ppb, with 5 hours exceeding the limit of 106 ppb (GB3095–2012, Grade I). The SO$_2$ mixing ratios exhibited an extremely significant downward trend ($-0.37$ ppb yr$^{-1}$, $R = -0.20$, $P < 0.01$) during the measurement period and a higher downward trend ($-1.10$ ppb yr$^{-1}$, $R = -0.22$, $P < 0.01$) from 2013 to 2017. The NO$_x$ mixing ratios exhibited a much smaller but significant downward trend ($-0.03$ ppb yr$^{-1}$, $R = -0.01$, $P < 0.05$). Details in the trends and the influencing factors will be discussed in Sect. 3.4.

As shown in Table 1, the average values $\pm 1\sigma$ (standard deviation) of SO$_2$, NO, NO$_2$, and NO$_x$ concentrations are 5.7 $\pm 8.4$ ppb, 1.1 $\pm 2.6$ ppb, 13.1 $\pm 10.9$ ppb, and 14.2 $\pm 12.4$ ppb, respectively. The results were close to the annual average concentrations of SO$_2$ (5.9 $\pm 10.0$ ppb), NO (0.8 $\pm 2.0$ ppb), NO$_2$ (13.8 $\pm 13.1$ ppb), and NO$_x$ (14.5 $\pm 14.0$

ppb) at SDZ in 2004 reported by Meng et al. (2009a). Compared with other background stations in China (Table 2),
the $SO_2$ and $NO_x$ mixing ratios at SDZ are both at a relatively high level.

**3.2 Monthly variations**

Surface $SO_2$ and $NO_x$ mixing ratios at SDZ showed a similar seasonal pattern with high values in winter and low
values in summer (Fig. 3). The highest $SO_2$ level appeared in winter (9.46 ppb) with the maximum monthly mean in
February (10.57 ppb), followed by that in spring (7.28 ppb) and autumn (5.01 ppb), and the lowest in summer (2.06
ppb) with the minimum in July (1.45 ppb). The concentration of $NO_x$ was higher in winter (18.12 ppb) and autumn
(16.51 ppb), lower in spring (12.95 ppb) and summer (9.24 ppb). The maximum monthly mean $NO_x$ appeared in
November (21.70 ppb) and the minimum one in August (8.69 ppb). The seasonal patterns of $SO_2$ and $NO_x$ at SDZ
were similar to those in urban and rural areas in North China (Meng et al., 2009b; Lin et al., 2012; Song et al., 2016;
Tang et al., 2016; Chen, 2017b; Zhao et al., 2020), which were characterized by high levels in the heating period and
low levels in summer.
The heating period in North China was from November to March. Coal burning was used to be the major source for
heating in the NCP, but it has been gradually substituted by natural gas since 2013 in urban areas. In the rural areas,
however, there was still burning of coal and wood for heating. The emissions of $SO_2$ and $NO_x$ in the heating period
were higher than those in the non-heating periods. Compared with the non-heating period, lower temperature, drier
air, weaker solar radiation, less precipitation, and lower mixing depth heights were found in the heating period,
resulting in lower atmospheric chemical reaction rate of $SO_2$ and $NO_x$, smaller removal effect of precipitation,
weaker vertical diffusion, longer atmospheric lifetime, and thus higher concentrations.

**3.3 Diurnal variations**

The average diurnal variations in $SO_2$ and $NO_x$ mixing ratios at SDZ in different seasons are shown in Fig. 4. The
$SO_2$ mixing ratios peaked at 11:00 in spring and summer, 14:00 in fall, and 21:00 in winter. The $NO_x$ mixing ratios
peaked at 1:00 in winter, 2:00 in spring, fall and summer. In addition, the valley of $SO_2$ diurnal cycle appeared at
5:00 in spring and summer, 6:00 in fall and winter, whereas for $NO_x$ it was at 12:00 in spring, 13:00 in summer,
13:00 in fall, and 12:00 in winter, respectively. The diurnal behaviors of $NO_x$ and $SO_2$ mixing ratios are different.
Generally, the average daily amplitudes of $SO_2$ are 3.0 ppb in spring, 2.0 ppb in summer, 4.4 ppb in fall, and 3.7 ppb
in winter, respectively, while the average daily amplitudes of $NO_x$ are 6.8 ppb in spring, 6.3 ppb in summer, 10.6
ppb in fall and 10.5 ppb in winter, respectively.

**3.4 Long-term trends of $SO_2$ and $NO_x$ mixing ratios**

Figure 5a shows the annual mean $SO_2$ mixing ratios from 2004 to 2016 at SDZ site, as well as the annual $SO_2$ emissions in North China (including Beijing, Tianjin, Hebei, Shanxi and Inner Mongolia). The annual mean $SO_2$ mixing ratio in 2004 was from Meng et al. (2009a). The $SO_2$ emission peaked in 2006 and then decreased with years. Meanwhile, the annual mean $SO_2$ mixing ratio reached a high level around 7.6 ppb during 2006-2008, and then began to decline thereafter. A rebound in $SO_2$ emission occurred in 2011, while a lagged rise of $SO_2$ mixing ratio occurred in 2012. Overall, the annual mean $SO_2$ exhibited a significant decreasing trend of $-0.36$ ppb $yr^{-1}$ ($-6.1$ % $yr^{-1}$, $R = -0.84$, $P < 0.01$) from 2004 to 2016 and a greater decreasing trend of $-0.56$ ppb $yr^{-1}$ ($-7.4$ % $yr^{-1}$, $R = -0.95$, $P < 0.01$) from 2008 to 2016.

Figure 5b shows the long-term variations in the annual 5th and 95th percentile values of the hourly mean $SO_2$ in different years. The 95th percentile indicated the influence of polluted air masses, while the 5th percentile indicated the influence of clean air masses. Similar to the trends of annual mean $SO_2$ mixing ratios, the 95th percentile of $SO_2$ reached its peak (30.87 ppb) in 2007, and a little decrease in 2008 (29.19 ppb). After 2008, it began to decline. Compared with the $SO_2$ level in 2008, there was a great decrease ($-19.8$ %) in 2009, but from 2009 to 2012 there was no significant decline in annual mean of $SO_2$. The most significant downward trend of the 95th percentile of $SO_2$ was found from 2012 to 2016 with a rate of $-3.98$ ppb $yr^{-1}$ ($-16.3$ % $yr^{-1}$, $R = -0.99$, $P < 0.01$). However, the 5th percentile of $SO_2$ mixing ratios did not change significantly of $-0.05$ ppb $yr^{-1}$ ($-2.6$ % $yr^{-1}$, $R = -0.15$, $P = 0.6$) from 2005 to 2016.

The annual mean of $NO_x$ shows an increasing trend of $+0.37$ ppb $yr^{-1}$ ($+3.4$ % $yr^{-1}$, $R = 0.38$, $P = 0.40$) from 2005 to 2010 with strong fluctuations (Fig. 5c,d). The annual $NO_x$ mean reached the peak value (16.93 ppb) in 2010, and exhibited a significant downward trend of $-0.77$ ppb $yr^{-1}$ ($-4.5$ % $yr^{-1}$, $R = 0.95$, $P < 0.01$) from 2010 to 2016 (Fig. 5c). The 95th percentile of the hourly mean of $NO_x$ firstly increased during 2005-2012 with $+0.02$ ppb $yr^{-1}$ ($+0.1$ % $yr^{-1}$, $R = 0.73$, $P < 0.05$) and then decreased during 2012–2016 with $-0.03$ ppb $yr^{-1}$ ($-4.7$ % $yr^{-1}$, $R = 0.95$, $P < 0.05$). Similar to $SO_2$, the annual 5th percentile of $NO_x$ mixing ratios did not change significantly ($-1.7$ % $yr^{-1}$, $R = -0.18$, $P = 0.58$) from 2005–2016 (Fig. 5d).

We regrouped $NO_x$ and $SO_2$ data into 4 subsets according to the heating period (November–March), spring (April–May), summer (June–August), and autumn (September–October). The long-term trends of the four subsets are shown in Fig. 6. The $SO_2$ mixing ratios showed significant downward trends of $-0.96$ ppb $yr^{-1}$ ($-8.0$ % $yr^{-1}$, $R = -0.99$, $P < 0.01$) in the heating period, $-0.39$ ppb $yr^{-1}$ ($-5.2$ % $yr^{-1}$, $R = -0.84$, $P < 0.01$) in spring, $-0.24$ ppb $yr^{-1}$

($-4.3$ % $yr^{-1}$, $R = -0.92$, $P < 0.01$) in autumn, and $-0.18$ ppb $yr^{-1}$ ($-7.7$ % $yr^{-1}$, $R = -0.87$, $P < 0.01$) in summer. The
large reduction in the $SO_2$ level in the heating period was largely related to burning natural gas instead of coal for
domestic heating (Qiu et al., 2017; Li et al., 2020).
Except for autumn, the trends of the annual mean $NO_x$ mixing ratios in other seasons showed a similar pattern that
$NO_x$ mixing ratio rose firstly and then declined significantly. The annual mean of $NO_x$ in autumn showed a
downward but statistically insignificant trend of $-0.08$ ppb $yr^{-1}$ ($-0.6$ % $yr^{-1}$, $R = -0.28$, $P = 0.38$) from 2005 to
2016. In other seasons, the peaks of $NO_x$ appeared in different years. The $NO_x$ mixing ratios showed significant
downward trends of $-1.16$ ppb $yr^{-1}$ ($-5.4$ % $yr^{-1}$, $R = -0.84$, $P < 0.05$) in the heating period during 2012–2016, $-1.07$
ppb $yr^{-1}$ ($-7.6$ % $yr^{-1}$, $R = -0.96$, $P < 0.01$) in spring during 2012–2017, and $-0.67$ ppb $yr^{-1}$ ($-4.5$ % $yr^{-1}$, $R = -0.87$,
$P = 0.01$) in summer during 2011–2016.
**4 Discussion**
**4.1 The influence of emission control on long-term trends of $NO_x$ and $SO_2$**
The annual mean and the 95th percentile of $SO_2$ mixing ratios at SDZ from 2004 to 2016 were significantly
correlated with the annual $SO_2$ emissions in North China with correlation coefficients of 0.85 ($P < 0.01$) and 0.88 ($P$
$< 0.01$), respectively. The decreasing rates of annual mean and 95th percentile of $SO_2$ mixing ratios from 2004 to
2016 at SDZ were $-6.1$ % $yr^{-1}$ and $-6.2$ % $yr^{-1}$, respectively, which were higher than the trend ($-3.1$ % $yr^{-1}$) of the
annual $SO_2$ emission in the NCP, but very close to the trend ($-6.3$ % $yr^{-1}$) of the annual $SO_2$ emission in Beijing.
This indicated that surface $SO_2$ mixing ratios at SDZ were more influenced by the emission in Beijing than other
provinces in the NCP.
There seemed a lag between the variation of $SO_2$ mixing ratios and the emissions (Fig. 5a,b; Fig. S1a,b) and surface
$SO_2$ mixing ratio in 2012 was evidently inconsistent with the emission trend, which indicated the complexity of the
effect of reducing $SO_2$ emission on $SO_2$ mixing ratios. The effectiveness and timing of pollution control policies, as
well as the change of meteorology year by year, would cause their asynchronous trends. China has implemented a
series of stringent clean air actions from 2013 to 2017, and the "*Beijing 2013-2017 Clean Air Action Plan*" was
the most comprehensive and systematic pollution control program in Beijing (UN Environment, 2019). Before
2013, there would be some emissions being not counted for some reasons by local government, as the change
in the 95% percentile of $SO_2$ mixing ratios did not show a similar decreasing trend of the mean $SO_2$ mixing
ratios from 2009 to 2011. Another reason would be the change in $SO_2$ mixing ratios at the SDZ regional
background site was not as obvious as the change in Beijing urban and other polluted areas as Lin et al. (2012)
had stated. Changes in meteorology would also lead to a decoupling of emissions and measured $SO_2$ and $NO_2$
values, but it cannot be quantified how much the changes contributed to this time shift.
Taking 2008 as the base year, a stronger decreasing trend of $-7.4$ % yr$^{-1}$ ($R = -0.95$, $P < 0.01$) from 2008 to 2016 for
the annual mean $SO_2$ mixing ratio can be found, as well as a significant decreasing rate of ($-4.5$ % yr$^{-1}$, $R = -0.81$, $P$
$< 0.01$) for the annual 5 % percentile of $SO_2$ mixing ratios. More strict emission control measures had been
implemented for the 2008 Beijing Olympic Games, where the $SO_2$ pollution control had long-term effects and
benefits as Lin et al. (2012) had pointed out. Surface $SO_2$ mixing ratios in Beijing in the first half year of 2008
before the Olympic game, held in August and September, showed higher values than that in the rest of the year (Lin
et al., 2012). We believe the higher emission before the Olympics was due to more activities in preparing the
Olympic game. Although more reduction in $SO_2$ was seen in the post-Olympics period, the $SO_2$ mixing ratio showed
a higher annual mean in 2008 than in 2009. Theoretically, the worldwide economic crisis in 2009 might cause a
lower level of $SO_2$ but considering the economic stimulation measures implemented by the government, we do not
think the economic crisis played a significant role. Moreover, the higher $NO_x$ emission in 2009 than in 2008
supports our view. The improvement of energy structure has been speeded up in Beijing from 2009, which might be
a more important factor. An assessment by the United Nations Environment Programme reported that the significant
decline in $SO_2$ mixing ratios and emissions from 1997–2017 was largely due to the $SO_2$ control measures in Beijing
and the surrounding areas, especially the transformation of coal-fired boilers, energy structure adjustments and the
end treatment of $SO_2$ tail gas (UN Environment, 2019). The $SO_2$ observation at SDZ background site confirmed the
effect of $SO_2$ reduction.
Before 2011, the annual mean $NO_x$ showed an increasing trend with fluctuation year by year. There is a steep
increase in $NO_x$ in 2010, as well as that in 2006. It is worth noting that the motor vehicles in Beijing in 2010 had
increased significantly from the previous year (see Figure S2), since the policy of purchase restriction in motor
vehicle was implemented in 2011. In addition, $NO_x$ emissions from power plants and industrial sources were not
strictly controlled before 2011. Therefore, more $NO_x$ would be emitted in years with prosperous economy.
According the analysis by Krotkov et al. (2016) and Duncan et al. (2016), $NO_2$ pollution over Northeast China has
reached its peak in 2011, and there have large decreases over Beijing, Shanghai, and the Pearl River Delta, which
were likely associated with local emission control efforts. Beijing has adopted the policy of "new car purchase
restriction" lottery number purchase since January 1, 2011 and has implemented the plan for further promoting the
elimination and renewal of old cars since August 1, 2011. New glass emission standard for air pollutants from the
flat glass industry (GB 26453-2011) was also implemented in this year. After 2010, the annual mean and 95th
percentile of $NO_x$ mixing ratios correlated significantly ($R = 0.94$, $P < 0.01$ and $R = 0.82$, $P < 0.05$, respectively)
with the annual $NO_x$ emission in North China, but the $NO_x$ mixing ratios exhibited more fluctuations than $NO_x$
emissions (Fig. 5c, 5d). As shown in Fig. S1c and S1d, the annual mean $NO_x$ mixing ratios were also significantly
correlated with the $NO_x$ emission in Beijing ($R = 0.93$, $P < 0.01$) from 2010 to 2016 (Fig. S1c). However, the 95th
percentile of $NO_x$ did not show a significant correlation ($R = 0.80$, $P = 0.06$) (Fig. S1d), indicating that high values
of $NO_x$ at SDZ may be much more affected by $NO_x$ emissions from other North China regions than Beijing. The
decreasing rates of $-4.8$ % $yr^{-1}$ ($R = -0.92$, $P < 0.01$) for the annual mean and $-4.5$ % $yr^{-1}$ ($R = -0.82$, $P < 0.05$) for
the 95th percentile $NO_x$ mixing ratios from 2011 to 2016 at SDZ were lower than the one ($-8.8$ % $yr^{-1}$, $R = -0.94$, $P$
$< 0.01$) for the annual $NO_x$ emission in the NCP and ($-9.0$ % $yr^{-1}$, $R = -0.96$, $P < 0.01$) in Beijing. Unlike the annual
mean or 95th percentile values, the 5th percentile of $NO_x$ mixing ratios from 2011 to 2016 did not exhibit a
significant trend ($-5.0$ % $yr^{-1}$, $R = -0.54$, $P = 0.27$) at SDZ.
It indicated that surface $NO_x$ mixing ratios at SDZ was relatively weakly influenced by the emission reduction in
Beijing and its surrounding areas in the NCP compared with the condition of $SO_2$, probably because there were
more emission sources for $NO_x$ than for $SO_2$. For example, although the coal-burning source pollution control
measures adopted in the *the Clean Air Action* have helped to reduce $NO_x$ emissions, the increase in the amount of
motor vehicles led to an increase in $NO_x$ emission from the traffic (Fontes et al., 2018; Sun et al., 2018; Zhang et al.,
2019; Zhang et al., 2020). In addition, the change of atmospheric transport conditions and the expansion of urban
scale may lead to the downward trend of $NO_x$, but not as obvious as that of $SO_2$ at SDZ (Lin et al., 2012).
Fortunately, $NO_x$ pollution control measures on coal-burning source and vehicle pollution had also begun to achieve
more significant outcome since 2011 (Krotkov et al., 2016; UN Environment, 2019). Especially, vehicle pollution
control was strengthened through the improvement of oil quality and promotion of new energy vehicles. As a result,
Beijing's motor vehicle growth rate decreased from 19.7 % in 2010 to 3.6 % in 2011 and the number of new energy
vehicles had an increase of 431 % from 2013 to 2016 (Figure S2).
**4.2 Variations in $NO_x$ and $SO_2$ mixing ratios in different periods**
We regrouped the $NO_x$ and $SO_2$ data into 4 subsets in 4 different time stages (Stage I: 2005–2008, Stage II:
2009–2012, Stage III: 2013–2014, and Stage IV: 2015–2017). Key pollution control measures had been
implemented in different stages, e.g., emission controls for the 2008 Beijing Olympic Games, *the State Council Air*
*Pollution Prevention and Control Action Plan (Action Plan 2013–2017)* and *Beijing 2013–2017 Clean Air Action*
*Plan*. Details of the pollution prevention plans and its implementation can be found in UN Environment (2019) and
in Zheng et al. (2018), in which, control process and specific measures for coal combustion and motor vehicles in
Beijing from 1998 to 2017, and China's clean air policies implemented during 2010–2017 had been reviewed. Since
2015, the government of Beijing–Tianjin–Hebei region has promoted the application of electric energy substitution
using electric energy instead of traditional fossil energy (Wang et al., 2020).
The average diurnal variations in $SO_2$ and $NO_x$ at SDZ in four stages are shown in Fig. 7 and Fig. S3. The $SO_2$ levels
and their amplitudes of the average diurnal variation continued to decrease as the stage time went by. The
differences in $SO_2$ among the 4 different periods are significant (P<0.001) from the One-Way ANOVA test, and the
differences between the two groups are also significant (P<0.01) from t-test. The diurnal amplitude of $SO_2$ was 4.16
ppb in Stage I and 0.94 ppb in Stage IV. The peak time of $SO_2$ in Stage IV appeared at 11:00 instead of the former
16:00. The peak value decreased significantly, from 9.38 ppb in Stage I to 3.19 ppb in Stage IV, with a factor of
−66.0 %. This phenomenon indicated that the control measures implemented in the period 2013–2017 have not only
had notable effects in reducing emissions from power plants, but also had significant achievement in the emission
control of non-electric industries such as industrial boilers and kilns (Zhang et al., 2019), which made the emission
intensity of $SO_2$ pollutants from elevated sources weaker than that in the Stage I.
Different from $SO_2$, the average diurnal of $NO_x$ mixing ratios did not show a gradual decrease over time and with
values of Stage II > Stage III > Stage I > Stage IV. For $NO_x$, the differences among the 4 different periods are
significant (P=0.01) from the One-Way ANOVA test, but the differences between the two groups are only significant
(P<0.01) between Y2009-2012 and Y2015-2017 from t-test. In addition, the diurnal variations and the diurnal
amplitude of $NO_x$ did not change much with the daily amplitudes being about 8.52 ppb. The peak and valley
appeared respectively at about 2:00 and at about 13:00 in 4 stages. The increase of $NO_x$ and the decreasing of $SO_2$ in
Stage II tells the fact of much more effective of pollution control measures on $SO_2$ rather than $NO_x$ implemented in
Beijing and other places. China intensified its acid rain control in the beginning of this century by much more strict
control of $SO_2$ emissions from coal-fired power plants. However, the control of $NO_x$ emissions remained weak until
the introduction of the new Emission Standard of Air Pollutants for Thermal Power Plants (GB13223-2011) (Wang
et al., 2019). Such major difference in $SO_2$ and $NO_x$ emission control caused an earlier peak for $SO_2$ (around 2006)
and a later peak for $NO_x$ (around 2011-2012) (Li et al., 2017). The emission data for North China (Figure 5) nearly
resemble the nationwide situation and the mixing ratio data at SDZ (Figure 7) are consistent with the general trends
of $SO_2$ and $NO_x$ emissions. At the same time, the amount of motor vehicles has been rapidly increasing, resulting in
an increase in $NO_x$ emissions from vehicle exhaust.
Figure 8 shows the rose maps of $SO_2$ and $NO_x$ mixing ratios in 4 different time periods (Figure S4 and S5 are rose
maps in different seasons, Table S1 is frequency distributions of wind directions in different stages). High $NO_x$
values were in broader wind sectors except NW–NNW–N–NNE, whereas high $SO_2$ values were mainly in
W–WSW–SW–SSW sectors. Except for the SSW sector, $SO_2$ mixing ratios in other wind directions showed a
decreasing trend over stages. Both the severely polluted areas and the slightly polluted areas have the same
characteristics of decreasing in $SO_2$ level over time (Table 3). Unlike the highest $SO_2$ mixing ratio being in Stage I
(2005–2008), the highest $NO_x$ mixing ratios was in Stage II (2009–2012). The overall levels of $SO_2$ and $NO_x$ in the
Stage IV reached the lowest values among the four stages. Compared with those at the stage with the highest mixing
ratios of $NO_x$ and $SO_2$, the reduction ranges in Stage IV are 52.2 %–76.4 % for $SO_2$ and 3.8 %–45.3 % for $NO_x$ in
different wind sectors. Much more reduction in $SO_2$ than $NO_x$ indicates that the electric energy substitution policy in
Beijing–Tianjin–Hebei region has been much more effective on $SO_2$ reduction than $NO_x$.
The $SO_2/NO_x$ ratio, obtained from the reduced major axis regression of the daily mean $SO_2$ and $NO_x$ mixing ratios,
exhibited a significant change from 0.84 during 2005–2008 to 0.30 during 2015–2017. The possible reason for this
phenomenon was that the control measures including the upgrading of end treatment facilities of coal-fired power
plants, the conversion of coal to clean energy, and the elimination of coal-fired boilers, which were taken in the early
stage of *the Clean Air Action*, had greatly reduced $SO_2$ emissions rather than $NO_x$. Another reason could be an
increase in the number of motor vehicles (Figure S2) and relatively more difficulties in emission control on the
mobile sources. Unlike emissions from industries, emissions from automobiles are relatively more difficult to
control. The reason that supports this argument is that emissions from industrial plants could be quantitatively
measured, thus control measures that require a reduction of a certain percentage in emissions could be implemented.
However, the estimation of emissions from automobiles bears large uncertainties in the first place. Though there are
also strict control regulations as to cars with license plates of a chosen number are not allowed to be on road on
certain days, the actual reduction in emissions also depends on the usage of other cars.
In the period of 2005–2012, the construction of new power plants and the amount of motor vehicle ownership
rapidly increased in the city. During this period, flue gas desulfurization devices have been widely used (Zhao et al.,
2008). However, the main management measures that required power plants to deploy denitrification devices for
reducing $NO_x$ emissions, have not been implemented until 2012, resulting in the increase of nitrogen oxide
emissions (Wang et al., 2010; Wang and Hao, 2012; Liu et al., 2016), and the contribution to the transport of $NO_x$ to
SDZ during this period.

**4.3 The different diurnal behaviors in $SO_2$ and $NO_x$ mixing ratios and their source origin**

The seasonal variations in $SO_2$ and $NO_x$ mixing ratios exhibited a similar pattern with high values in winter and low values in summer, and their daily mean values had a significant correlation ($R = 0.59$, $P < 0.01$). However, the diurnal variations in $SO_2$ and $NO_x$ mixing ratios were greatly different from each other (Figure 9). Due to the increased emissions, lower mixing depth and slower chemical conversions in winter, $SO_2$ values showed significant diurnal behavior in winter which was different from other seasons. Except for the winter, the $SO_2$ mixing ratios were higher during the daytime and lower during the nighttime in all seasons, while the $NO_x$ mixing ratios showed an opposite pattern. The different diurnal behaviors in $SO_2$ and $NO_x$ at SDZ might indicate a different origin of $SO_2$ and $NO_x$.

Due to the diurnal variation in the boundary layer, the mixing depth is higher during the daytime and the convective mixing is strong, which is conducive to the dilution and diffusion of pollutants. The photochemical reaction during the daytime is also conducive to the chemical transformation of pollutants. At night, the pollutants are easy to accumulate because of lower mixing depth and no photochemical processes. Therefore, the concentration of primary pollutants exhibits higher values during the nighttime and lower during the daytime in general. But the situation for $SO_2$ at SDZ was different. The higher $SO_2$ mixing ratios during the daytime suggested two possible mechanisms: (1) an elevated level of $SO_2$ aloft, which could be mixed downward to the ground due to the evolution of atmospheric boundary layer, causing higher ground-level $SO_2$ concentrations in the daytime. (2) upwind $SO_2$ sources and transport of plumes in the daytime.

Since the SDZ station is selected as WMO/GAW regional station, local anthropogenic emissions are well avoided. As SDZ is located on the north side of a valley with a northeast-southwest orientation, its dominant wind directions were from southwest and northeast with regular changes in diurnal wind directions (Figure S6). The southwest mouth of the valley is open to the NCP, so it is easily influenced by the air masses from the south polluted areas, like urban Beijing. As a result, the concentration rose maps of pollutants exhibited higher values in the southwest sectors than other sectors (Lin et al., 2008; Meng et al., 2009a). If only due to the influence by advection transport, the diurnal variations in $SO_2$ and $NO_x$ at SDZ should be similar. However, the two show obvious differences. The higher $SO_2$ mixing ratios during the daytime indicates an elevated level of $SO_2$ in a high air layer, which can be exchanged to the surface under the evolution of atmospheric boundary layer, causing a higher $SO_2$ value in the daytime. The 'unusual' phenomenon of the diurnal change in $SO_2$ has been noticed and explained by studies (Lin et al., 2008; Chen et al., 2009; Xu et al., 2014), but it lacked direct vertical profile measurements to support this

explanation. The daytime peak of $SO_2$ was not only found at SDZ, but also at different sites in urban and rural areas
in North China (Lin et al., 2012) and in the background area of the Yangtze River Delta (Qi et al., 2012). This may
be related to the fact that $SO_2$ is mainly emitted from elevated sources (Lin et al., 2012; Xu et al., 2014). The daily
maximum of $SO_2$ concentrations was caused by the downward mixing of $SO_2$ emitted by elevated sources in this
region. As strict and effective control measures were continuously implemented, the contribution from such a source
largely decreased and finally became negligible. Governed by the development of the planetary boundary layer, the
diurnal variation of $SO_2$ concentrations would peak around noon. This may be the cause of the shift in time of the
$SO_2$ maximum as mentioned in Section 4.2. Xu et al. (2014) have discussed the implications of this $SO_2$
noontime-peak phenomenon in sulphur deposition and transformation. At night, prevail north wind transported clean
air to SDZ. This process should be the major cause of the decreasing $SO_2$ levels during nighttime, since surface $SO_2$
mixing ratios depend on vertical air exchange. Enhanced relative humidity during nighttime should be also a loss
effect since $SO_2$ is a very soluble gas. In addition, dry deposition of $SO_2$ in a shallow nocturnal boundary layer
might lower the $SO_2$ level as well.
It can be seen that the $NO_x$ mixing ratios began to rise around noontime when the mixing depth was still elevating
(Figure 9). Obviously, $NO_x$ was affected by the transport of pollutants in the southern polluted area during the
noontime when the WD changed into southwest wind (Figure S6). Of course, motor vehicles running on the roads
and dispersing human activities can emit $NO_x$ as well as the transport from the south. As seen in Figure 8, the $NO_x$
rose map showed wider source origins than $SO_2$. However, $SO_2$ maintained a relatively high value instead of
increasing significantly, indicating that $SO_2$ mixing ratios were still mainly affected by downward mixing of
$SO_2$-richer air.

**Conclusion**

Measurements of surface $NO_x$ and $SO_2$ mixing ratios at SDZ regional atmospheric background site in the North
China Plain from the period 2005–2017, together with ancillary data, were summarized and used to study their
long-term trends and influencing factors. The average values $\pm 1\sigma$ (standard deviation) of $SO_2$ and $NO_x$ mixing
ratios were $5.7 \pm 8.4$ ppb and $14.2 \pm 12.4$ ppb, respectively. The seasonal variation in $SO_2$ and $NO_x$ at SDZ showed a
similar pattern with high values in winter and low values in summer, but the diurnal variations in $SO_2$ and $NO_x$
mixing ratio exhibited large differences in all seasons. The $SO_2$ mixing ratios were higher during the daytime and
lower during the nighttime, while the $NO_x$ mixing ratios showed higher values during the nighttime and lower
during the daytime. The different diurnal behaviors in $SO_2$ and $NO_x$ at SDZ indicated a different origin of $SO_2$ and
$NO_x$.
Overall, the annual mean $SO_2$ exhibited a significant decreasing trend of $-0.36$ ppb yr$^{-1}$ ($-6.1$ % yr$^{-1}$, $R = -0.84$, $P <$
$0.01$) from 2004 to 2016 and a greater decreasing trend of $-0.56$ ppb yr$^{-1}$ ($-7.4$ % yr$^{-1}$, $R = -0.95$, $P < 0.01$) from
2008 to 2016. The decreasing rates of annual mean and 95th percentile of $SO_2$ mixing ratios from 2004 to 2016 at
SDZ are very close to the one ($-6.3$ % yr$^{-1}$) of the annual $SO_2$ emission in Beijing. The annual mean of $NO_x$ showed
a fluctuating rise of $+0.37$ ppb yr$^{-1}$ ($+3.4$ % yr$^{-1}$, $R = 0.38$, $P = 0.40$) from 2005 to 2010, reaching the peak value
(16.93 ppb) in 2010, and then exhibited an extremely significant fluctuating downward trend of $-0.77$ ppb yr$^{-1}$
($-4.5$ % yr$^{-1}$. $R = 0.95$, $P < 0.01$) from 2010 to 2016. After 2010, the annual mean and 95 % percentile of $NO_x$
mixing ratios correlated significantly ($R = 0.94$, $P < 0.01$ and $R = 0.82$, $P < 0.05$, respectively) with the annual $NO_x$
emission in North China. The decreasing rates of $-4.8$ % yr$^{-1}$ ($R = -0.92$, $P < 0.01$) for the annual mean and $-4.5$ %
yr$^{-1}$ ($R = -0.82$, $P < 0.05$) for the 95th percentile $NO_x$ mixing ratios from 2011 to 2016 at SDZ are lower than the
one ($-8.8$ % yr$^{-1}$, $R = -0.94$, $P < 0.01$) for the annual $NO_x$ emission in the NCP and ($-9.0$ % yr$^{-1}$, $R = -0.96$, $P <$
$0.01$) in Beijing. It indicated that surface $NO_x$ mixing ratios at SDZ had a weaker response to the emission reduction
in Beijing and its surrounding areas in NCP than $SO_2$. The increase in the amount of motor vehicles and the weak
effectiveness of traffic restrictions have caused motor vehicle emissions on $NO_x$.
***Data availability.*** The data in this study can be publicly accessed via https://doi.org/10.7910/DVN/YFVLHV (Liu et
al., 2022).
***Author contributions.*** XL wrote the paper, WL developed the idea, formulated the research goals, and edited the
paper. LR, XX and ZQ edited the paper. WL, FD, DH, LZ, QS and YW carried out the measurement of $NO_x$ and
$SO_2$, and analysed the meteorological data.
***Competing interests.*** The authors declare that they have no conflict of interest.
**Acknowledgements.** This study was funded by the National Natural Science Foundation of China (Grant No.
91744206) and the Open Fund of Shangdianzi Atmospheric Background Station (SDZ2020615).

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

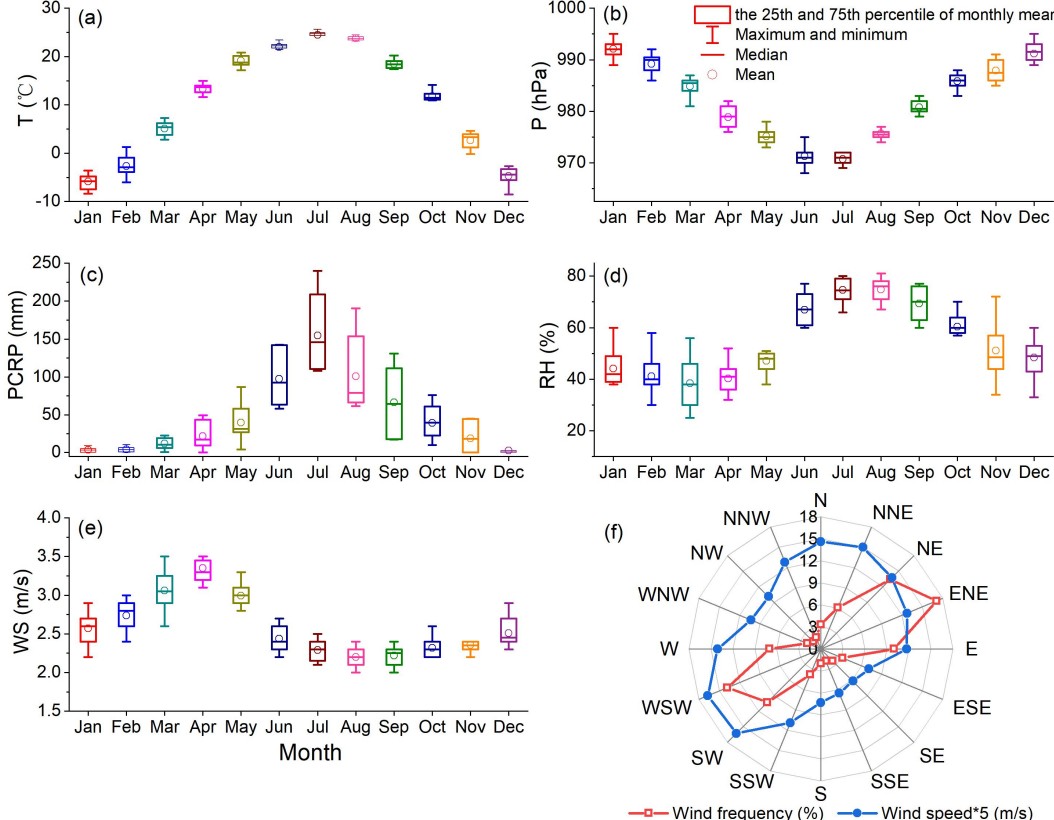


**Figure 1.** Monthly variations in **(a)** air temperature. **(b)** atmospheric pressure. **(c)** precipitation. **(d)** relative humidity. **(e)** wind speed. **(f)** wind rose map. at SDZ.

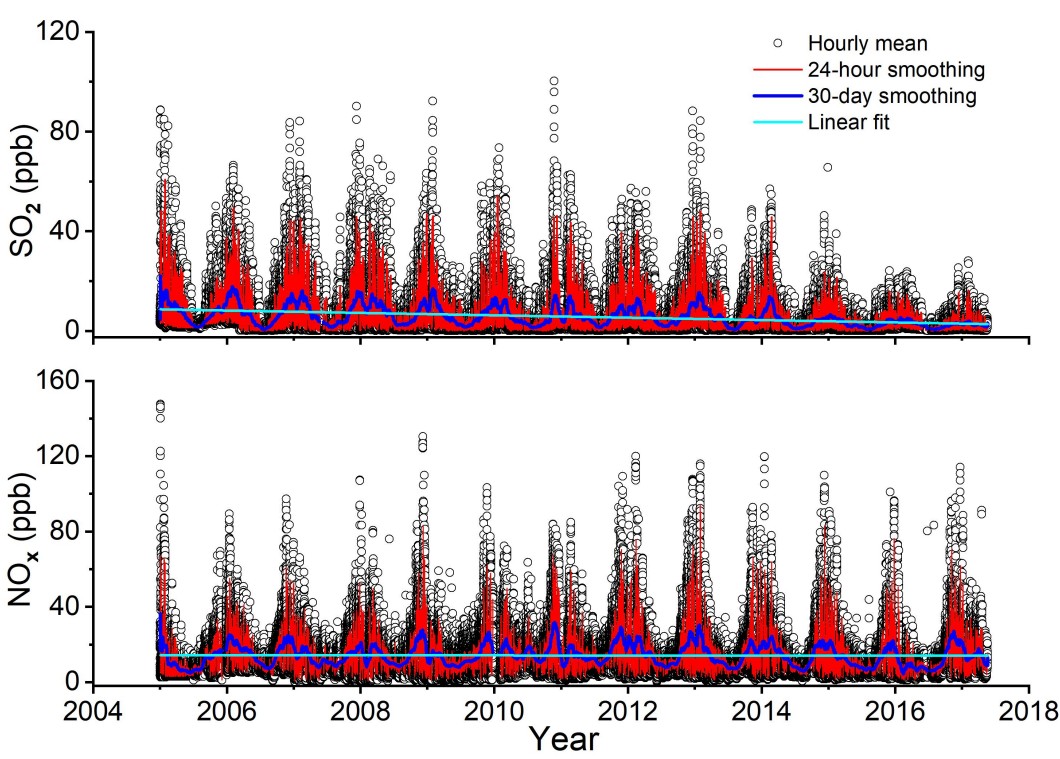


**Figure 2.** The time series variations in $SO_2$ and $NO_x$ mixing ratios at SDZ.

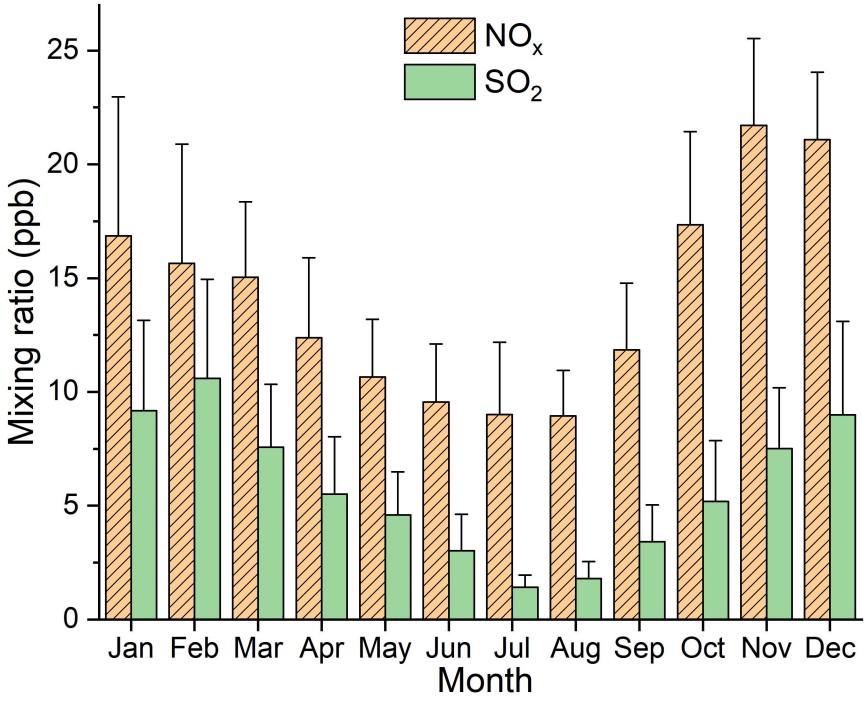


**Figure 3.** The average monthly mean of $SO_2$ and $NO_x$ mixing ratios with 1 σ at SDZ.

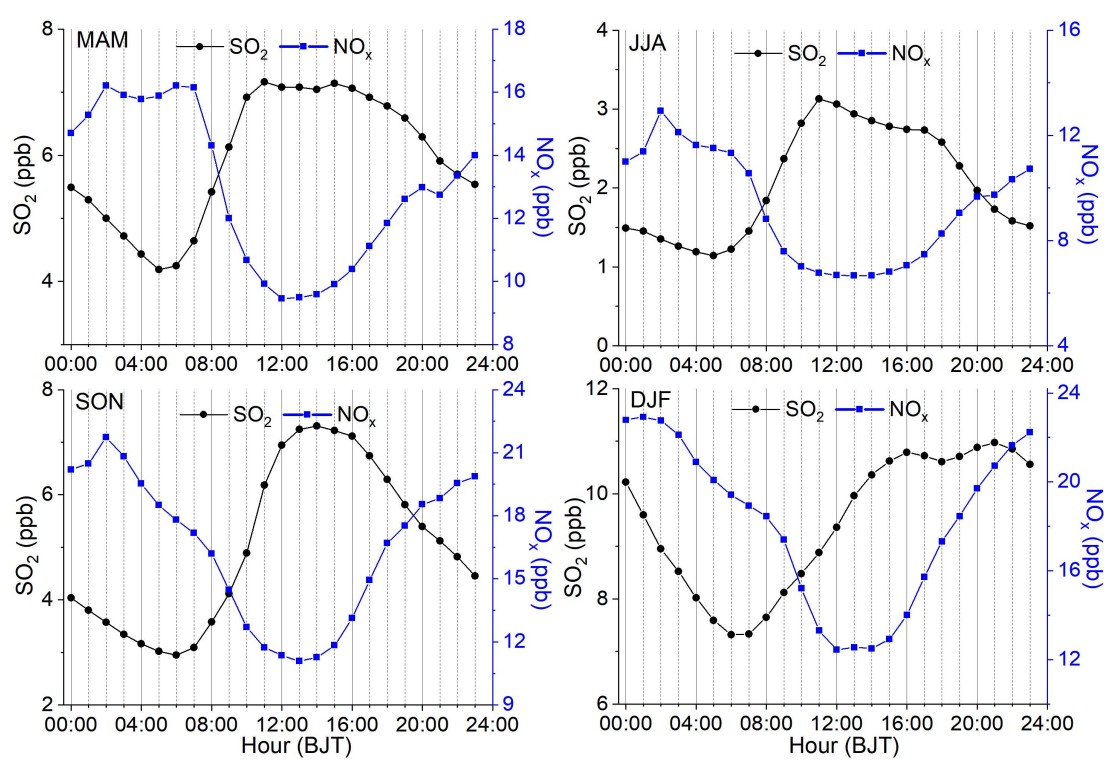


**Figure 4.** The Average diurnal variations in $SO_2$ and $NO_x$ mixing ratios in four seasons at SDZ.

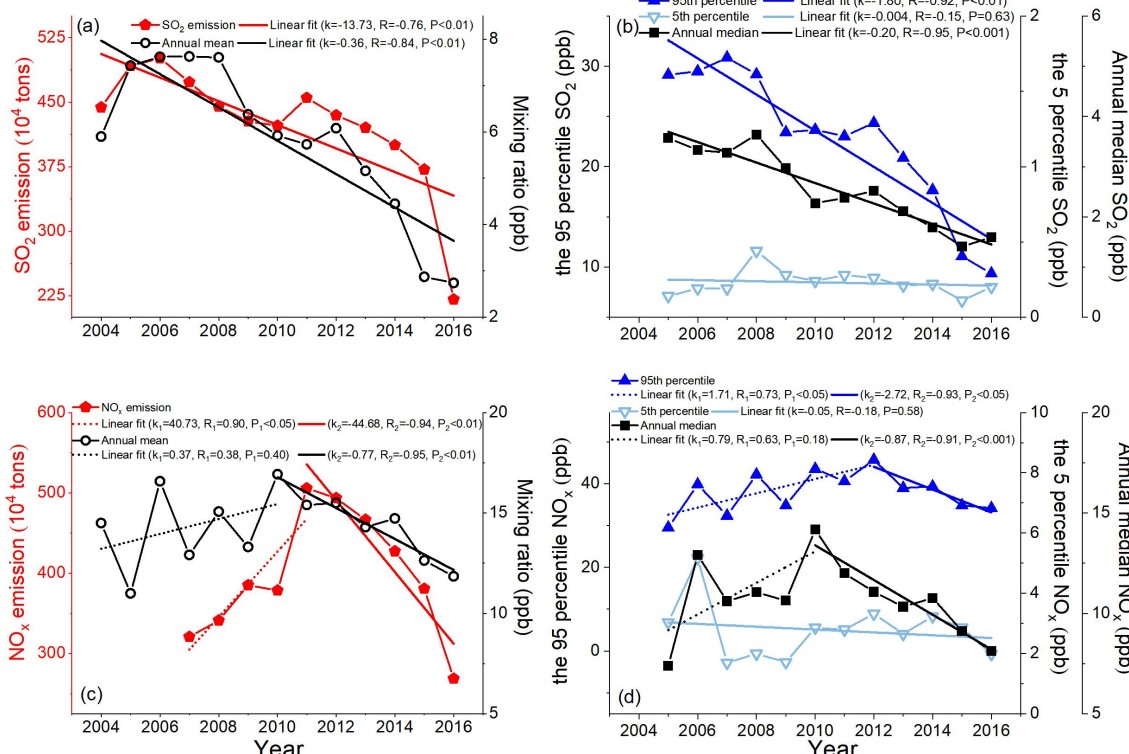


**Figure 5.** Annually variations in **(a)** $SO_2$ mixing ratios at SDZ and total $SO_2$ emissions in North China; **(b)** the 5th and 95th percentile of the hourly mean and annual median of $SO_2$ mixing ratios and $SO_2$ emissions in North China; **(c)** $NO_x$ mixing ratios at SDZ and total $NO_x$ emissions in North China; **(d)** the 5th and 95th percentile of the hourly mean and annual median of $NO_x$ mixing ratios and $NO_x$ emissions in North China. The emission data are from the 2005–2017 Yearbook of National Bureau of Statistics of China and China Statistical Yearbook on Environment provided by Ministry of Ecology and Environment of the People's Republic of China.

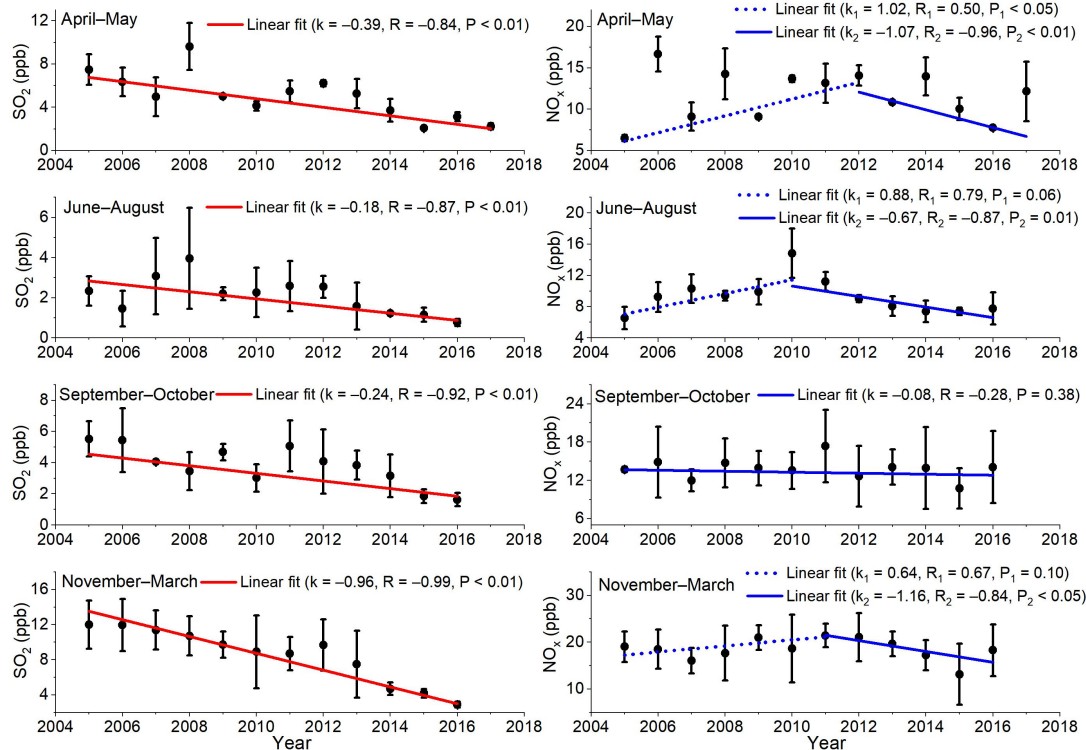


**Figure 6.** Long-term variations in monthly mean $SO_2$ and $NO_x$ mixing ratios with ± 1σ in different periods at SDZ. Heating period (November–March next year), spring (April–May), summer (June–August), and autumn (September–October).

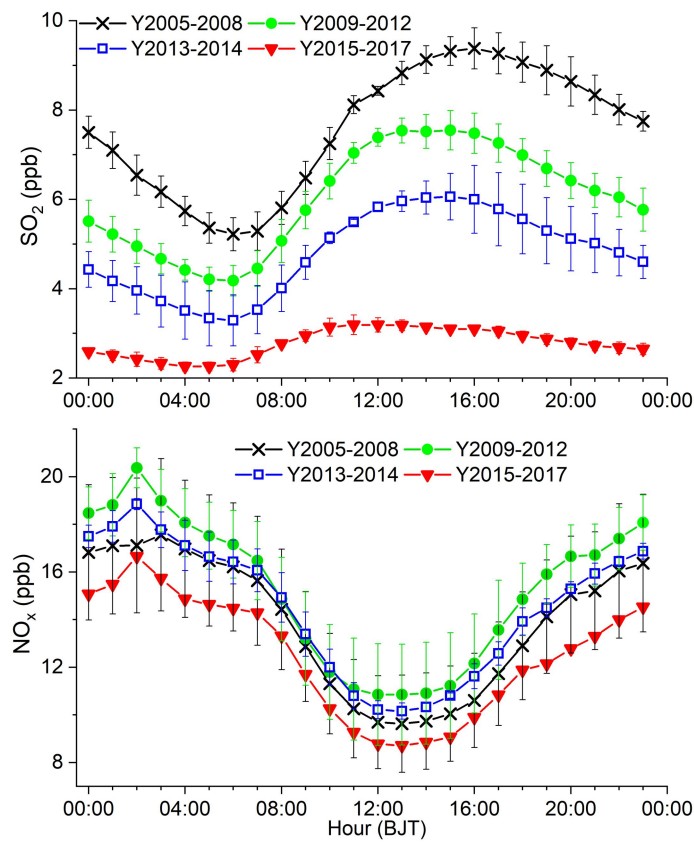


**Figure 7.** The average diurnal variations in $SO_2$ and $NO_x$ mixing ratios in 4 different stages at SDZ.

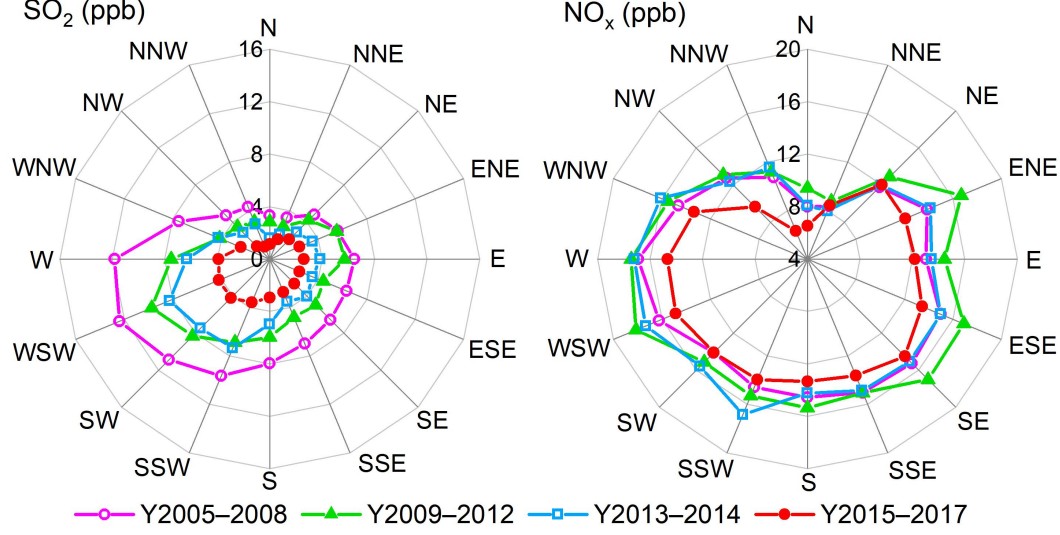

**Figure 8.** Mixing ratios of SO$_2$ and NO$_x$ during different stages as a function of wind direction at SDZ.

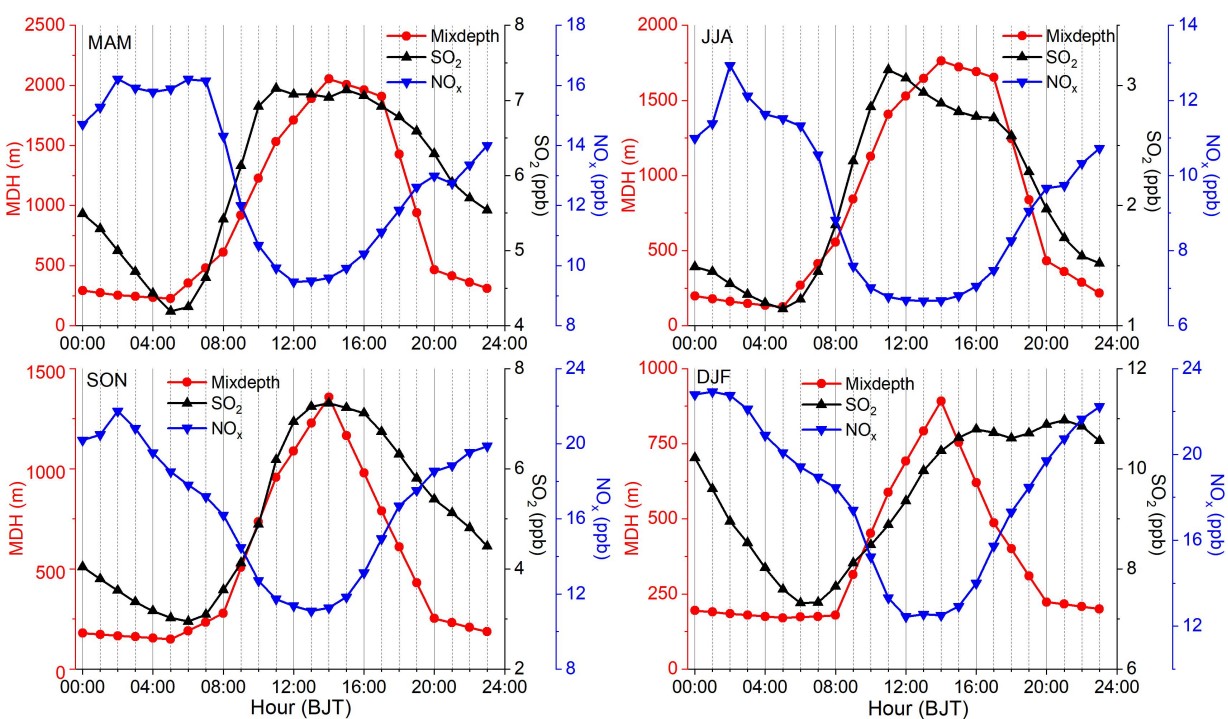

**Figure 9.** Diurnal variations in mixing depths in four seasons at SDZ.

**Table 1.** Statistics in the hourly mean of $SO_2$ and $NO_x$ mixing ratios at SDZ.

| | NO (ppb) | $NO_2$ (ppb) | $NO_x$ (ppb) | $SO_2$ (ppb) |
|---|---|---|---|---|
| Mean | 1.10 | 13.08 | 14.18 | 5.71 |
| Standard deviation | 2.58 | 10.89 | 12.36 | 8.44 |
| Median | 0.33 | 9.98 | 10.59 | 2.45 |
| Maximum | 83.34 | 124.41 | 147.58 | 100.34 |
| Minimum | 0.01 | 0.01 | 0.14 | 0.01 |
| Count number | 104923 | 104923 | 104923 | 105374 |


**Table 2.** $NO_x$ and $SO_2$ levels in the atmospheric background stations in China.

| Site | Time | $NO_x$ (ppb) | $SO_2$ (ppb) | References |
|---|---|---|---|---|
| SDZ(North China) | 2005.1–2017.5 | 14.2 ± 12.4 | 5.7 ± 8.4 | This study |
| Xinglong(North China) | 2005.5–2015.1 | – | 7.5 | (Bai et al., 2015) |
| Linan(Yangtze River Delta) | 2005.8–2006.7 | – | 11.1 ± 10.6 | (Qi et al., 2012) |
| | 2006.1–2016.12 | 13.6 ± 1.2 | 7.0 ± 4.2 | (Yin et al., 2022) |
| Wuyishan(East China) | 2011.3–2012.2 | 2.70 | 1.48 | (Su et al., 2013) |
| Dinghushan(South China) | 2009.1–2010.12 | 13.6 | 6.5 | (Chen, 2012) |
| Changbaishan(Northeast China) | 2009.1–2010.12 | 4.7 | 2.1 | (Chen, 2012) |
| Fukang(Northwest China) | 2009.1–2010.12 | 8.3 | 2.2 | (Chen, 2012) |
| Gonggashan(Southwest China) | 2017.1–2017.12 | 0.90 | 0.19 | (Cheng et al., 2019) |
| Jinsha(Central China) | 2006.6–2007.7 | 5.6 ± 5.5 | 2.8 ± 5.5 | (Lin et al., 2011b) |


**Table 3.** Trends of the hourly mean of the three sectors with the highest $SO_2$ level, the hourly mean of the three sectors with the lowest $SO_2$ level and their difference.

| | Highest $SO_2$ values (ppb) | Lowest $SO_2$ values (ppb) | Difference (ppb) |
|---|---|---|---|
| | W–WSW–SW–SSW sectors | NNW–N–NNE–NE sectors | |
| Y2005–2008 | 11.18 | 3.96 | 7.23 |
| Y2009–2012 | 8.12 | 3.20 | 4.91 |
| Y2013–2014 | 7.36 | 2.35 | 5.01 |
| Y2015–2017 | 3.95 | 1.48 | 2.48 |
