# Peer review of "Measurement report: Variations in surface $SO_2$ and $NO_x$ mixing ratios from 2004 to 2016 at a background site in the North China Plain"

_Atmospheric Chemistry and Physics, 2021_

## Author Comment (AC1)

We thank for the constructive comments and suggestions. We revised our manuscript according to the comments and suggestions. The following list the point-to-point response to the comments. The changed texts were highlighted with yellow color.

**Response to comments by referee # 1**

General comments:

This paper reports on NOx and $SO_2$ measurements at a background site in the North China Plain. The site, the instrumental setup, quality control and the data processing procedures have been described in detail. Data are compared to other data from other measurement sites. The long-term trend of both $SO_2$ and NOx, their diurnal and seasonal behavior are discussed and compared to emission data. As publications of long time series of NOx and $SO_2$ are rare, the manuscript should be published after these questions have been answered:

Specific comments:

1.   Line 66 and following

Here the authors describe the setup of the instrument. For NOx detection, a chemiluminescence analyzer has been used. Unfortunately, it is not stated how $NO_2$ is converted to NO for detection. Has a thermal or a photolytical converter been used? If a thermal converter has been used this would mean that a large fraction of the NOx would be in fact NOy, as nitric acid and nitrates would cause significant interferences in the $NO_2$ channel (Jung et al., 2017, Steinbacher et al., 2007). This interference should be discussed. How frequent was the conversion efficiency determined? Was an $NO_2$ gas standard used or was it done by gas phase titration?

Response: Thanks. The commercial NOx analyzer (Model 42CTL) uses a molybdenum $NO_2$-to-NO converter heated to about 325°C. We have also noticed the drawback of this technique, but have to accept what has been available since 2004. A favorable and direct $NO_2$ measurement technique based on cavity ring-down principle could be a choice in the future, but there would be another challenge in the availability of stable and reliable reference standard of $NO_2$. We discussed the possible interference in another paper (see https://doi.org/10.5194/acp-22-1015-2022). We cited the discussion in the revised paper. The following text are added in the revised paper.

"In Model 42C-TL NOx analyzer, $NO_2$ is converted to NO by a molybdenum $NO_2$-to-NO converter heated to about 325°C. The conversion efficiency was checked annually using gas phase

titration of an NO standard with $O_3$. The converter was replaced if the conversion efficiency was found lower than 96%. The drawback in this $NO_2$ converter was known to suffer from the interference of other NOy compounds such as PAN and $HNO_3$ (Steinbacher et al., 2007; Jung et al., 2017), which was also discussed in Yin et al. (2022). As it is not possible in our case to remove the interference, the reported $NO_2$ and NOx levels should be treated as upper limits."

We added this information in the revised paper, see Page 3, Line 65-70.

Steinbacher, M., Zellweger, C., Schwarzenbach, B., Bugmann, S., Buchmann, B., Ordóñez, C., Prevot, A. S. H., and Hueglin, C.: Nitrogen oxide measurements at rural sites in Switzerland: Bias of conventional measurement techniques, J. Geophys. Res-atmos., 112(D11), https://doi.org/10.1029/2006JD007971, 2007.

Jung, J., Lee, J., Kim, B., and Oh, S.: Seasonal variations in the $NO_2$ artifact from chemiluminescence measurements with a molybdenum converter at a suburban site in Korea (downwind of the Asian continental outflow) during 2015–2016, Atmos. Environ., 165, 290-300, https://doi.org/10.1016/j.atmosenv.2017.07.010, 2017.

Yin, Q., Ma, Q., Lin, W.*, Xu, X., and Yao, J.: Measurement report: Long-term variations in surface NOx and $SO_2$ from 2006 to 2016 at a background site in the Yangtze River Delta region, China, Atmos. Chem. Phys., 22, 1015–1033, 2022. https://doi.org/10.5194/acp-22-1015-2022.

How was mixing depth determined?

Response: We used a hybrid single-particle Lagrangian integrated trajectory model (Hysplit4.9) from National Oceanic and Atmospheric Administration, USA, with the NCEP–NCAR reanalysis meteorological data set (https://ready.arl.noaa.gov/archives.php) to calculate the atmospheric mixed layer heights.

We added this information in the revised paper, see Page 4, Line 79-81.

2. Line 162 and following

The authors state that there is a lag between $SO_2$ mixing ratios and emissions. What can cause such a lag? Change in meteorology would only lead to a decoupling of emissions and measured $SO_2$ and $NO_2$ values but not to a time shift.

Response: Yes, there seemed a lag between the variation in mean and the 95% percentile of $SO_2$ mixing ratios and the emissions. Surface $SO_2$ mixing ratio in 2012 was evidently inconsistent with the emission trend. China has implemented a series of stringent clean air actions from 2013 to 2017, and the "Beijing 2013-2017 Clean Air Action Plan" was the most comprehensive and systematic pollution control program in Beijing (UN Environment, 2019). Before 2013, there would be some emissions being not counted for some reasons by local government, as the change in the 95% percentile of $SO_2$ mixing ratios did not show a similar decreasing trend of the mean $SO_2$ mixing ratios from 2009 to 2011. Another reason would be the change in $SO_2$ mixing ratios at the SDZ regional background site was not as obvious as the change in Beijing urban and other polluted areas as Lin et al. (2012) had stated. Changes in meteorology would also lead to a decoupling of emissions and measured $SO_2$ and $NO_2$ values, but it cannot be quantified how much the changes contributed to this time shift.

We add the information in the revised paper, see Page 7, Line 170-181.

3. Line 166 and following

The authors write that the 2008 Olympic games should affect the emissions. However, in figure 5 the measured $SO_2$ values are highest in 2008 and low in 2009. Is it possible the effect of the worldwide economic crisis in 2009 is larger than the effect of the Olympic games and not well represented in the emission data?

Response: In Lin et al. (2012), we indicated that surface $SO_2$ mixing ratios in Beijing in the first half year of 2008 before the Olympic game, held in August and September, showed higher values than that in the rest of the year. We believe the higher emission before the Olympics was due to more activities in preparing the Olympic game. Although more reduction in $SO_2$ was seen in the post-Olympics period, the $SO_2$ mixing ratio showed a higher annual mean in 2008 than in 2009. Theoretically, the worldwide economic crisis in 2009 might cause a lower level of $SO_2$ but considering the economic stimulation measures implemented by the government, we do not think

the economic crisis played a significant role. Moreover, the higher NOx emission in 2009 than in 2008 supports our view. The improvement of energy structure has been speeded up in Beijing from 2009, which might be a more important factor.

We added information in the revised paper, see Page 7-8, Line 186-194.

4. Line 246 and following

In chapter 4.3. the authors explain the different diurnal features of $SO_2$ and NOx. The profile of NOx coincides with the change in mixing depth while the diurnal profile of $SO_2$ is opposite to it. The profile of $SO_2$ is explained by transport from $SO_2$-rich air from above which originate from remote, not necessarily elevated, sources, which will increase $SO_2$ concentration during daytime. However, during nighttime an efficient loss process must reduce the $SO_2$ concentration again. What is the loss process that reduces the concentration of $SO_2$ reduced during nighttime?

Response: Due to the typical mountain-plain topography, south wind prevails in the daytime and north wind prevails at night in the North China Plain (Lin et al., 2008), similar diurnal wind directions like that in Figure S6. South wind brings polluted air mass and north wind transports clean air to SDZ. This process should be the major cause of the decreasing $SO_2$ levels during nighttime, since surface $SO_2$ mixing ratios depend on vertical air exchange. Enhanced relative humidity during nighttime should be also a loss effect since $SO_2$ is a very soluble gas. In addition, dry deposition of $SO_2$ in a shallow nocturnal boundary layer might lower the $SO_2$ level as well.

We added information in the revised paper, see Page 11-12, Line 298-304.

5. Line 246 and following

With respect to the daily profile of NOx it was argued that it is the result of transport processes during noontime. But isn´t it more likely that NOx is emitted from local sources close to the ground, as motor vehicles and small burners This was observed at the background site at Linan (Yin et al., 2022), which showed similar diurnal cycles and similar mean NOx concentrations.

Response: Since the SDZ station is selected as WMO/GAW regional station, local anthropogenic emissions are well avoided. The station is situated on the south slope of a hill, on the north hill side of a valley with a northeast-southwest orientation. The southwest mouth of the valley is open to Beijing and the south plain. In the valley, motor vehicles running on the roads and dispersing

human activities can emit NOx as well as the transport from the south. As seen in Figure 8, the NOx rose map showed wider source origins than SO₂. It seems that the diurnal NOx cycle is not only similar with that at Linan, but also with those in urban areas, which might tell us a truth of regional characteristics of air pollution in eastern China. But these should be carefully studied in the future.

We add information in the revised paper, see Page 12, Line 307-309.

**Technical corrections:**

1.   Line 91: High values in winter and low values in summer

Response: Accepted.

2.   Line 102: lower mixing depth heights

Response: Accepted.

3.   Line 112: 6.27 ppb (ppb is missing)

Response: Accepted.

4.   Line 112: I think a one-digit precision of the values is sufficient here

Response: Accepted.

5.   Line 238: More difficulties

Response: Accepted.

6.   Line 247: high values in winter and low values

Response: Accepted.

7.   Line 249: greatly different from each other

Response: Accepted.

8.   Line 287: low values in

Response: Accepted.

9.   Line 288: exhibited great differences

Response: Accepted.

---

## Author Comment (AC2)

**Response to comments by referee # 2**

General comments:

The study by Liu et al. reports long term measurements of surface $SO_2$ and NOx from a background measurement site in North China. I consider the data suited for publication in ACP, the manuscript is well organized and well written. However, concerning the data analysis, presentation and interpretation I have a few concerns detailed in the specific comments below that should be accounted in the revised version. Please consider making your data publicly available.

We appreciate the referee's valuable comments. We have addressed each comment as below and revised the manuscript accordingly. We have also made our data publicly available as described in "Data availability: The data in this study can be publicly accessed via https://doi.org/10.7910/DVN/YFVLHV (Liu et al., 2022)."

**Specific comments:**

**1. l8-9: Are the differences in diurnal cycle of $SO_2$ and $NO_X$ due to sources or chemistry?**

Response: The differences in diurnal cycle of $SO_2$ and NOx were attributed to emissions and chemical and meteorological processes. However, the major reason for the differences in all seasons in this study should be ascribed to apparent different emission sources of these two trace gases and meteorological effects on their concentrations. The diurnal cycle of NOx, with higher values during night and lower ones during daytime, can easily be explained by accumulation of NOx in the shallow nocturnal boundary layer and dilution and chemical removal during daytime. The diurnal cycle of $SO_2$, with a clear peak around noon, seems to be less understandable. However, $SO_2$ is mainly emitted by power plant stacks and other coal-burning sources (normally with higher stacks), unlike NOx, which is mainly emitted by vehicles and other lower height sources. The elevated emission height makes big difference in the diurnal cycle of $SO_2$ under the influences of boundary layer evolution, transport and other processes. According the study by Xu et al. (2014), the $SO_2$ noontime-peak phenomenon could be attributed to four processes, i.e., down-mixing, plume transport, mountain valley breeze and fog/severe haze. For the SDZ site, mountain valley breeze is most important, followed by down-mixing.

Xu, W., Zhao, C., Ran, L., Lin, W., Yan, P., and Xu, X.: $SO_2$ noontime-peak phenomenon in the North China Plain, Atmos. Chem. Phys., 14, 7757-7768, https://doi.org/10.5194/acp-14-7757-2014, 2014.

**2. l32-34: Please explicitly state the difference between regional and local/suburban pollution.**

Response: Local/suburban pollution is closely associated air pollutants emitted locally. The pollution is limited on a smaller scale such as a town, a city or an urban area. Primary air pollutants exhibited a diurnal variation mainly following the diurnal pattern of emissions and containing the signature of diurnal differences of vertical mixing and chemical consumption. Regional pollution occurs over the whole region and is usually associated with large-scale

emissions and significantly influenced by transport and accompanying processes, such as chemical reactions, deposition, etc. In China, city clusters have been formed for decades, air pollution often shows regional characteristics.

We have added some information in Line 35-39.

**3. l36: What are "typical regional characteristics"? Please provide this information.**

Response: The typical regional characteristics of $SO_2$ as observed at different types of sites in this region in previous studies was a seasonal maximum in winter and minimum in summer and a daytime peak throughout the year. The typical regional characteristics of NOx was a seasonal maximum in winter and minimum in summer and a daytime valley in the diurnal pattern.

We have added some information in Line 42-43.

**4. l43: Please provide examples for the few studies (references).**

Response: We have added a reference as below.

Bai, J. H., Wu, Y. M., Chai, W. H., Wang, P. C., Wang, G. C.: Long-term variation of trace gases and particulate matter at an atmospheric background station in North China, Advances in Geosciences, 5, 248-263, http://dx.doi.org/10.12677/ag.2015.53025, 2015.

**5. l98 & l100-104: Does "heating season" mean winter? Please disentangle the season = winter and the cause for the high NOx values = heating. What is mainly used for heating in China? Coal, wood, oil, gas, electricity? Are your conclusion that the high NOx and $SO_2$ concentrations are caused by higher emissions in combination with the meteorological conditions that slow down the reduction by less transport and slower chemical reactions?**

Response: As mentioned in the manuscript, the heating season in North China was from November to March (officially from 15 November to 15 March). So, "heating season" is not an accurate expression. We have changed it to "heating period". During the heating period, a large amount of air pollutants such as $SO_2$, NOx, and particles are released from burners of centralized and domestic heating facilities. Before 2013 coal was dominantly burned for heating in the NCP, but it has been gradually substituted by natural gas since then in urban areas. In rural areas, however, burning of coal or wood has been a common practice for heating. The high levels of NOx and $SO_2$ were partly due to strong emissions during the heating period and partly due to meteorological conditions that favored the accumulation of air pollutants due to less transport and slower chemical processes.

We have added some information in Line 115-119.

**6. l106-113: What causes the diurnal cycles? Please explain.**

Response: The diurnal cycle of $SO_2$, showing a daytime peak, was mainly resulted from mountain valley breeze and downward mixing of $SO_2$ emitted by elevated sources (Xu et al., 2014). The

diurnal cycle of NOx, showing a daytime valley, was mostly due to the active photochemical reactions and vertical mixing during the day and accumulation in the shallow nocturnal boundary layer. We discussed the causes in Section 4.3 and only described the result here.

Xu, W., Zhao, C., Ran, L., Lin, W., Yan, P., and Xu, X.: $SO_2$ noontime-peak phenomenon in the North China Plain, Atmos. Chem. Phys., 14, 7757-7768, https://doi.org/10.5194/acp-14-7757-2014, 2014.

**7. l141-145: Could the significant peak of $SO_2$ concentrations in 2008 be correlated with the (preparations for) the Olympic summer games 2008?**

Response: Control measures have been gradually taken a couple of years before the Olympic Games Particularly, very strict measures started from July 2008. The concentration of $SO_2$ during September-October and November-March in 2008 was actually lower than that in previous years as a result of the effective control measures. A higher $SO_2$ in the early half year of 2008 was observed, and the higher emission before the Olympics is believed to be due to more activities in preparing the Olympic game (Lin et al., 2012).

Lin, W., Xu, X., Ma, Z., Zhao, H., Liu, X., and Wang, Y.: Characteristics and recent trends of sulfur dioxide at urban, rural, and background sites in North China: Effectiveness of control measures, J. Environ. Sci., 24, 34-49, https://doi.org/10.1016/s1001-0742(11)60727-4, 2012.

**8. l146-152: Please explain the cause for the changes in NOx.**

Response: Here we just presented the changing result of NOx and we discussed the causes in Section 4.1.

**9. Fig 6: Which months are shown for November-March? Jan, Feb, Mar, Nov, Dec of e.g 2008, or Nov 2007 - Mar 2008, or Nov 2008 - Mar 2009?**

Response: The period November-March 2008 was November 2008 to March 2009.

**10. Fig 6b: The splitted fits should be consistently covering the same time periods: 2005-2010 and 2010-1017. What caused the change in NOx trends 2010/2011?**

Response: Thanks for the suggestion. We refitted the plots by splitting the same time periods: 2005-2010 and 2010-1017, and the result is shown in Figure R1. For April-May, the new split presents an insignificant trend from the old one. Since the peaks of seasonal NOx appeared in different years, we kept the previous splits in order to find the significant downward trend years.

[Figure]

**Figure R1. Figure 6b (left) and the revised Figure 6b (right).**

According the analysis by Krotkov et al. (2016) and Duncan et al. (2016), $NO_2$ pollution over Northeast China has reached its peak in 2011, and there have large decreases over Beijing, Shanghai, and the Pearl River Delta, which were likely associated with local emission control efforts. Since January 1, 2011, Beijing has adopted the policy of "new car purchase restriction" lottery number purchase. Since August 1, 2011, Beijing has implemented the plan for further promoting the elimination and renewal of old cars. New glass emission standard for air pollutants from the flat glass industry (GB 26453-2011) was also implemented in this year.

We discussed it in Section 4.1, see Line 213-218.

Krotkov, N. A., McLinden, C. A., Li, C., Lamsal, L. N., Celarier, E. A., Marchenko, S. V., Swartz, W. H., Bucsela, E. J., Joiner, J., Duncan, B. N., Boersma, K. F., Veefkind, J. P., Levelt, P. F., Fioletov, V. E., Dickerson, R. R., He, H., Lu, Z., and Streets, D. G.: Aura OMI observations of regional $SO_2$ and $NO_2$ pollution changes from 2005 to 2015, Atmos. Chem. Phys., 16, 4605-4629, https://doi.org/10.5194/acp-16-4605-2016, 2016.

Duncan, B. N., Lamsal, L. N., Thompson, A. M., Yoshida, Y., Lu, Z., Streets, D. G., Hurwitz, M. M. and Pickering, K. E.: A space-based, high-resolution view of notable changes in urban NOx pollution around the world (2005-2014), J. Geophys. Res. Atmos., 121(2), 976-996, https://doi.org/10.1002/2015JD024121, 2016.

**11. 6b April-May: Why is the fit put through the lowest values and does not seem to account for the high values in 2006, 2008, 2010, 2012, 2014, 2017? I'm not fully convinced by the increasing and then decreasing trend. Do the median values support these trends?**

Response: We checked the fitting curve and the median values fit similar trend (R). We fitted the curve not only based on the mean (median) values but also considering their uncertainties as weighting factors.

[Figure]

**Figure R2.** Long-term variations in **(a)** monthly mean and **(b)** monthly median NOx mixing ratios with 1-sigma for April–May in different year at SDZ.

**12. l169: Which day or month did the emission control measures for the Olympic games came into effect? Shouldn't this be visible in the seasonal data in Fig. 6?**

Response: Strict emission control measures took effect in July 2008. It can be seen in Fig. 6 that $SO_2$ concentrations exhibited a large variability during June-August in 2008, also $SO_2$ concentrations during the two periods of September-October and November-March were apparently lower than that in previous years. Lin et al. (2012) had discussed the changes in $SO_2$ between the pre-Olympic and the post-Olympic periods.

**13. Fig 5 right column: Please add data for the median. Maybe the NOx median fluctuates less than the NOx mean? Does the median also show a trend over time? Please add a description of the left and right y-axis to the figure caption. I assume that the 95 percentile refers to the left axis and the 5 percentile to the right axis. Is this correct?**

Response: We have added data for the median in Fig. 5 right column. The NOx median shows trends similar to the NOx mean. We added a description of the left and right y-axis to the figure caption.

[Figure]

**Figure 5.** Annually variations in **(a)** $SO_2$ mixing ratios at SDZ and total $SO_2$ emissions in North China; **(b)** the 5th and 95th percentile of the hourly mean and annual median of $SO_2$ mixing ratios and $SO_2$ emissions in North China; **(c)** $NO_x$ mixing ratios at SDZ and total $NO_x$ emissions in North China; **(d)** the 5th and 95th percentile of the hourly mean and annual median of $NO_x$ mixing ratios and $NO_x$ emissions in North China. The emission data are from the 2005–2017 Yearbook of National Bureau of Statistics of China and China Statistical Yearbook on Environment provided by Ministry of Ecology and Environment of the People's Republic of China.

**14. Fig5c: What caused the steep increase in NOx in 2010?**

Response: Before 2011, the annual mean NOx showed an increasing trend with fluctuation year by year. We also have no clues about this steep increase in NOx in 2010, as well as that in 2006. It is worth noting that the motor vehicles in Beijing in 2010 had increased significantly from the previous year (see Figure S2), since the policy of purchase restriction in motor vehicle was implemented in 2011. In addition, NOx emissions from power plants and industrial sources were not strictly controlled before 2011. Therefore, more NOx would be emitted in years with prosperous economy.

We added information in the revised manuscript, see Line 208-212.

**15. l201: Please provide information on the details of the pollution prevention plans, especially with respect to SO₂ and NOx, so that the reader can see how strict the measures were and how they changed over time.**

Response: Details of the pollution prevention plans and its implementation can be found in UN Environment (2019), A REVIEW OF 20 YEARS' Air Pollution Control in Beijing (http://sthjj.beijing.gov.cn/bjhrb/resource/cms/2019/04/2019041916301550241.pdf) and in Zheng et al. (2018). Control process and specific measures for coal combustion and motor vehicles in Beijing from 1998 to 2017, and China's clean air policies implemented during 2010–2017 had been reviewed. We added information in the revised paper, see Line 247-249.

UN Environment. A Review of 20 Years' Air Pollution Control in Beijing. United Nations Environment Programme, Nairobi, Kenya, 2019.
Zheng, B., Tong, D., Li, M., Liu, F., Hong, C., Geng, G., Li, H., Li, X., Peng, L., Qi, J., Yan, L., Zhang, Y., Zhao, H., Zheng, Y., He, K., and Zhang, Q.: Trends in China's anthropogenic emissions since 2010 as the consequence of clean air actions, Atmos. Chem. Phys., 18, 14095-14111, https://doi.org/10.5194/acp-18-14095-2018, 2018.

**16. l207-208: Please add what caused the shift in time of the maximum and what are the implications?**

Response: The daily maximum of SO₂ concentrations was caused by the downward mixing of SO₂ emitted by elevated sources in this region. As strict and effective control measures were continuously implemented, the contribution from such a source largely decreased and finally became negligible. Governed by the development of the planetary boundary layer, the diurnal variation of SO₂ concentrations would peak around noon. Xu et al. (2014) have discussed the implications. We discussed it in Section 4.3, see Line 337-347.

Xu, W., Zhao, C., Ran, L., Lin, W., Yan, P., and Xu, X.: SO₂ noontime-peak phenomenon in the North China Plain, Atmos. Chem. Phys., 14, 7757-7768, https://doi.org/10.5194/acp-14-7757-2014, 2014.

**17. l217-220: Please show in how far the changes in NOx correlate with the emission control measures in Beijing and/or NCP. I think this is speculative and giving details on the pollution prevention plans could support your conclusion.**

Response: Here, we draw this conclusion based on the opposite trends of the observational NOx and SO₂ mixing ratios. The increase of NOx and the decreasing of SO₂ in Stage II tells the fact of much more effective of pollution control measures on SO₂ rather than NOx. China intensified its acid rain control in the beginning of this century by much more strict control of SO₂ emissions from coal-fired power plants. However, the control of NOx emissions remained weak until the introduction of the new Emission Standard of Air Pollutants for Thermal Power Plants (GB13223-2011) (Wang et al., 2019). Such major difference in SO₂ and NOx emission control caused an earlier peak for SO₂ (around 2006) and a later peak for NOx (around 2011-2012) (Li et al., 2017). Our emission data for North China (Figure 5) nearly resemble the nationwide situation

and the mixing ratio data at SDZ (Figure 7) are consistent with the general trends of $SO_2$ and NOx emissions.

We add information in the revised manuscript, see Line 267-275.

Wang, N., Lyu, X., Deng, X., Huang, X., Jiang, F., Ding, A.: Aggravating $O_3$ pollution due to NOx emission control in eastern China, Science of the Total Environment, 677, 732-744, https://doi.org/10.1016/j.scitotenv.2019.04.388, 2019.

Li, M., Liu, H., Geng, G., Hong, C., Liu, F., Song, Y., Tong, D., Zheng, B., Cui, H., Man, H., Zhang, Q. and He, K.: Anthropogenic emission inventories in China: a review, National Science Review, 4, 834–866, https://doi.org/10.1093/nsr/nwx150, 2017.

**18. Fig7.: Do you have an explanation for the small peak at 2 am? Are the differences in $SO_2$ and NOx between the years significant? Please provide uncertainty range, e.g. 1-sigma-range for $SO_2$ and NOx.**

**Again, what are the uncertainty ranges? I assume, that the decreasing trend for $SO_2$ is robust. However, I'm not convinced that the differences in NOx between the different periods are outside the 1-sigma range.**

Response: Frankly, we do not have any clues about the small peak at 2 am yet.

In this section, time intervals were divided according to the average diurnal change of $SO_2$ in different years (see Figure S3 in supplementary material). Although the difference in diurnal change of NOx was not as obvious as $SO_2$ (see Figure R3 here), we applied the same time intervals here.

Figure R4 shows the result of adding 1-sigma ranges for $SO_2$ and NOx in Figure 7. For $SO_2$, the differences among the 4 different periods are significant (P<0.001) from the One-Way ANOVA test, and the differences between the two groups are also significant (P<0.01) from t-test. For NOx, the differences among the 4 different periods are significant (P=0.01) from the One-Way ANOVA test, but the differences between the two groups are only significant (P<0.01) between Y2009-2012 and Y2015-2017 from t-test.

We added information in the revised paper, see Line 253-255 and Line 263-265.

[Figure]

**Figure R3.** The average diurnal variations in NOx mixing ratios in different years at SDZ.

[Figure]

**Figure R4.** 1-sigma ranges for SO$_2$ and NOx are added in Figure 7.

**19. l238: Please detail what you mean with "relatively more difficult in emission control". Are there less regulations with respect to traffic emission control? Are the traffic emission control regulation less strictly adhered to?**

Response: Unlike emissions from industries, emissions from automobiles are relatively more difficult to control. The reason that supports this argument is that emissions from industrial plants could be quantitatively measured, thus control measures that require a reduction of a certain percentage in emissions could be implemented. However, the estimation of emissions from automobiles bears large uncertainties in the first place. Though there are also strict control

regulations as to cars with license plates of a chosen number are not allowed to be on road on certain days, the actual reduction in emissions also depends on the usage of other cars.

We added information in the revised manuscript, to see Line 294-299.

**20. l246-278: When discussing the differences in the diurnal cycle of SO2 and NOx their different lifetimes also should be considered. With SO2 lifetimes of ~13-10h in summer and 48-58h in winter [e.g. 1] transport from the NCP to the measurement site certainly has an impact. However, NOx lifetimes are shorter and significantly differ between summer (~6h) and winter (~24h) [e.g. 2] and day (29h) and nighttime (6h) [e.g. 3]. Here transport not always may have an impact and the observed NOx levels might be more local. Please discuss. Moreover, is there a diurnal cycle in the local emissions of SO$_2$ and NOx?**

**References**

**[1] Lee, C., Martin, R. V., van Donkelaar, A., Lee, H., Dickerson, R. R., Hains, J. C., Krotkov, N., Richter, A., Vinnikov, K., and Schwab, J. J.: SO2 emissions and lifetimes: Estimates from inverse modeling using in situ and global, space-based (SCIAMACHY and OMI) observations, J. Geophys. Res., 116, D06304, https://doi.org/10.1029/2010JD014758, 2011.**

**[2] Shah, V., Jacob, D. J., Li, K., Silvern, R. F., Zhai, S., Liu, M., Lin, J., and Zhang, Q.: Effect of changing NOx lifetime on the seasonality and long-term trends of satellite-observed tropospheric NO2 columns over China, Atmos. Chem. Phys., 20, 1483–1495, https://doi.org/10.5194/acp-20-1483-2020, 2020.**

**[3] Kenagy, H. S., Sparks, T. L., Ebben, C. J., Wooldrige, P. J., Lopez-Hilfiker, F. D., Lee, B. H., et al. (2018). NOx lifetime and NOy partitioning during WINTER. Journal of Geophysical Research: Atmospheres, 123, 9813–9827, https://doi.org/10.1029/2018JD028736.**

Response: We thank the referee for this helpful comment. We agree with the referee that a longer lifetime may be more associated with an impact from transport on a regional scale. Since the SDZ station is selected as WMO/GAW regional station, local anthropogenic emissions are well avoided. The station is situated on the south slope of a hill, on the north hill side of a valley with a northeast-southwest orientation. The southwest mouth of the valley is open to Beijing and the south plain. In the valley, motor vehicles running on the roads and dispersing human activities can emit NOx as well as the transport from the south. As seen in Figure 8, the NOx rose map showed wider source origins than SO$_2$. It seems that the diurnal NOx cycle is not only similar with that at Linan, but also with those in urban areas, which might tell us a truth of regional characteristics of air pollution in eastern China. But these should be carefully studied in the future.

**Technical corrections:**

**1. l8: and other instances: valley -> minimum**

We have revised the manuscript accordingly.

**2. l16: What is SDZ? Please introduce abbreviations before first usage.**

We have revised "at SDZ" as "at the Shangdianzi (SDZ) background site".

**3. l91: high value -> high concentrations**

We have revised the manuscript accordingly.

**4. l115,123,132,: showed -> shows**

We have revised the manuscript accordingly.

**5. l205: were -> are**

We have revised the manuscript accordingly.

**6. l221: "is" -> "shows" or "are"**

We have revised the manuscript accordingly.

**7. l232: has -> has been**

We have revised the manuscript accordingly.

**8. l237: should -> could**

We have revised the manuscript accordingly.

**9. l243: were not be > have not been**

We have revised the manuscript accordingly.

**10. l247: high -> high concentrations**

**low -> low concentrations**

We have revised the manuscript accordingly.

**11. l272: Please specify what "YRD" is.**

We have revise "the YRD" as "the Yangtze River Delta".

**12. l277: was -> were**

We have revised the manuscript accordingly.

**13. l285: exhibited greatly different for all > exhibited large differences in all**

We have revised the manuscript accordingly.

**14. l299-300: "... had weaker influence than SO2 by the emission reduction... " -> had a weaker response to the emission reduction in ... than SO2**

We have revised the manuscript accordingly.

---

## Author Comment (AC3)

**Response to comments by community # 1**

**1.What causes the difference in the diurnal variation of NOx and SO$_2$?**

Response: The differences in diurnal cycle of SO$_2$ and NOx were attributed to emissions, meteorological and chemical processes. However, the major reason for the differences in all seasons in this study should be ascribed to apparent different emission sources of these two trace gases.